# Behavior of Soft Tissue around Platform-Switched Implants and Non-Platform-Switched Implants: A Comparative Three-Year Clinical Study

**DOI:** 10.3390/jcm10132955

**Published:** 2021-06-30

**Authors:** Davide Farronato, Mattia Manfredini, Marco Farronato, Pietro Mario Pasini, Andrea Alain Orsina, Diego Lops

**Affiliations:** 1Department of Medicine and Surgery, School of Dentistry, University of Insubria, 21100 Varese, Italy; davide@farronato.it (D.F.); pm.pasini@gmail.com (P.M.P.); aa.orsina@gmail.com (A.A.O.); 2Department of Oral Surgery, School of Dentistry, University of Milan, 20122 Milan, Italy; 3Department of Orthodontics, School of Dentistry, University of Milan, 20122 Milan, Italy; marco.farronato@unimi.it; 4Department of Prosthodontics, School of Dentistry, University of Milan, 20142 Milan, Italy; diego.lops@unimi.it

**Keywords:** dental implants, tissue engineering, platform-switching, implant connection, soft tissue maturation

## Abstract

To verify the influence of platform-switching (PS) on soft tissue behavior by comparing the soft tissue stability around implants with and without PS, during three years of follow-up. The study included patients treated with fixed dentures supported by implants with an internal connection. The radiographic distance between the first bone-to-implant contact (FBIC) and the implant shoulder was assessed. Additionally, the presence of keratinized facial mucosa and the prosthetic crown height (TH) were monitored for three years from the delivery of the definitive crown. These parameters were measured for two different groups: platform-switched implants in the PS group and non-platform-switched (NPS) implants in the NPS group. Seventy-seven implants were considered in the statistical analysis. After three years, the overall FBIC mean value was 0.31 ± 1.00 mm. However, the mean FBIC was 0.66 ± 0.97 mm for the NPS group and −0.05 ± 0.91 mm for the PS group. Moreover, a mean recession of 0.54 ± 1.39 mm was measured for the NPS group, whereas a mean coronal migration of 0.17 ± 0.95 mm was measured for the PS group. A significant correlation was also found between the presence of PS and ΔTH (p ≤ 0.01) over the three years of follow-up. The absence or presence of platform-switching would appear to affect the tendency of the gingival buccal margin towards recession or creeping. Additionally, implant-abutment platform-switching seems to help prevent peri-implant soft tissue recession over time when compared to implants without PS.

## 1. Introduction

At present, fulfilling the esthetic expectations of patients is one of the most challenging tasks in implant dentistry [1] as, recently, attention has moved to long-term esthetic outcomes [2]. As a result, the success criteria in implantology have changed, according to the evolution of implant design and patient requests [3]. Especially for single implant restorations in the anterior area, the esthetic outcome plays a fundamental role and is a need for the patient [1]. Thus, an implant, which does not integrate perfectly in the patient’s oral cavity from the esthetic point of view, can no longer be considered satisfactory. Certainly, the position of the implant, as well as the quantity and quality of the hard and soft tissues, and their adaptation over time are all factors that condition the esthetic result of an implant-supported rehabilitation [4]. Araujo and colleagues [5] reported that important changes can impact hard and soft tissue following dental extractions and continue until two years after the implant insertion and prosthetic rehabilitation [4,5]. These variations are related to vascular support reduction and the loss of mechanical function [4,6]. In particular, soft tissue needs to adapt to the conditions of the local setting [4,7]. The potential causes of esthetic implant failures have previously been analyzed by Buser, including anatomic factors, such as horizontal or vertical bone deficiencies, and iatrogenic factors, such as the wrong implant selection or the incorrect implant positioning, where all were noted as critical for an esthetic implant-supported restoration [8]. Meanwhile, Wang investigated the soft tissue thickness around implant-supported restorations and the impact of thin and thick soft tissue biotypes [9]. The findings confirmed the soft tissue biotype as an important parameter in esthetic implant restoration, improving immediate implant success and preventing future mucosal recession [9]. Another factor influencing the soft tissue stability around dental implants over time is the prosthetic emergence profile [10]. In particular, the proper prosthetic contour and maintenance of the right amount of tissue were considered key factors [10]. Essentially, the esthetic outcome of an implant-supported restoration seems to be strictly related to the buccal hard- and soft-tissue height [1]. The effect of platform-switching (Figure 1), highlighted by Lazzara [11], is currently well-established in the literature [12]. Dental implants incorporating platform-switching (PS) preserve the crestal bone around the top of the implant [13,14,15,16,17] and alter the starting point from which crestal bone remodeling occurs [11]. Following the crestal bone position over time has also shown that the soft tissue stability is also affected by the presence of platform-switching [18]. When analyzing the peri-implant soft tissue with PS implants, Tarnow found that PS seemed to help in preserving the ridge dimensions and enhancing the peri-implant soft tissue stability [18]. Accordingly, the present study attempted to verify the influence of PS on soft tissue, by comparing the soft tissue stability around implants with and without PS, during three years of follow-up after the definitive crown placement. Recession or creeping was measured by referring to the visible length of the prosthetic crown (Tooth Height, or TH) and its changes over time (Δ Tooth Height, or ΔTH).

## 2. Materials and Methods

### 2.1. Patient Selection 

The present prospective study was conducted in accordance with the fundamental principles of the 1975 Helsinki Declaration on clinical analysis involving human subjects (revised in 2008), and was additionally approved by the ethics committee of Insubria University (#826-0034086) “Studies on the survival and the surgical-prosthetic success of dental implants: Influence of the implant-abutment connection”. Ninety patients referred for implant rehabilitation of partially edentulous ridges from October 2011 to March 2014 were identified as candidates for dental implant treatment by referring to clinical and radiographic examination.

The inclusion criteria were: single or multiple mandibular and maxillary partial edentulism with the requirement for implant rehabilitation and age ≥ 18 years at the time of surgery. The exclusion criteria were: poor oral hygiene (full-mouth plaque score FMPS > 25% and full-mouth bleeding score FMBS > 25%), the abuse of drugs or alcohol, the existence of acute oral infections, remote or recent radiation therapy in the oro-maxillofacial area, recent chemotherapy, and pregnancy. Other exclusion criteria included the absence of facial keratinized mucosa, the need for soft tissue grafting at any stage of the treatment, the need for post-extractive implant placement, and therefore the presence of buccal and lingual/palatal residual bone of at least 1.5 mm at implant placement, that may influence the result in terms of soft tissue maturation [19]. Moreover, any implant showing signs of mucositis or periimplantitis over time was excluded.

### 2.2. Surgical Protocol

#### 2.2.1. Patient Evaluation 

The patients were clinically and radiographically appraised to design the proper treatment plan. Panoramic and perio-apical imaging was used as the first-level exam to evaluate the site before implant placement. In the case of suspected bone deficiency, a Cone-Beam CT was performed as a second level exam to assess the alveolar ridge width. The oral hygiene, stability of the remaining teeth, soft tissue health, amount of keratinized gingiva, stability of the remaining teeth, and other factors possibly influencing the treatment plan were all clinically estimated. 

Patients presenting posterior maxillary edentulism with a bone ridge height <8 mm were treated with a sinus lift and a heterologous bone graft. Except for sinus augmentation procedures, all other cases requiring a GBR technique were eliminated from the study. 

#### 2.2.2. Pharmacological Treatment

Antibiotic prophylaxis of amoxicillin + clavulanic acid 2 g/day for six days (Augmentin, GlaxoSmithKline plc., Brentford, UK), starting with 2 g 1 h before surgery or clindamycin 600 mg/day for six days (Dalacin, Pfizer Inc., New York, NY, USA) in penicillin-allergic patients was prescribed to reduce the possibility of infections. Anti-inflammatory therapy with NSAIDs was also recommended: ketoprofen lysine 80 mg/day for three days (OKI, Dompè, Shanghai, China), starting with 80 mg 1 h before surgery. Local anesthesia of lidocaine + adrenalin 1:50,000 (Ecocain, Molteni Dental Srl, Milan, Italy) was administered at the site of intervention and articaine + adrenalin 1: 100,000 (Citocartin, Molteni Dental Srl, Milan, Italy) at other sites.

#### 2.2.3. Surgery

A two-stage surgical technique was used for the implant placement. A full-thickness mucoperiosteal flap was elevated to expose the underlying alveolar bone and allow implant placement in the correct prosthetically driven position. The optimal implant site was prepared at 800 rpm according to the company indications (MegaGen AnyRidge, MegaGen Implant Co. Ltd., Daegu, Korea). The inter-implant distance was at least 3 mm, while an interproximal space of 1.5 to 3 mm was observed between an implant and an adjacent tooth [2]. The implants were positioned in the bleeding socket with hand pressure and driven to their final position using a dedicated mounter at 60 rpm. The implants were all placed 0.5 mm sub-crestally, referring to the buccal bone, as recommended by the manufacturer. A torque value between 30 and 60 Nm was accepted, depending on the bone density. The repositioned mucoperiosteal flap was then secured with a tension-free suture for primary healing and to avoid bacterial contamination. All of the implants were placed in native crestal bone. No implant was located in a regenerated or post-extractive site. 

#### 2.2.4. Post-Op Instructions

All patients observed a 10-day liquid diet, plus 15 days of soft foods following the implant placement. Oral hygiene included the modified Bass brushing technique and 30-s 0.2% chlorhexidine (Curasept, Curasept SpA, Saronno, Italy) rinses three times a day for one week after surgery. The patients were recommended not to use removable partial dentures that may apply pressure over the surgery site. In some cases, an adhesive provisional Maryland bridge was applied, taking care not to touch the tissue over the implant.

### 2.3. Prosthetic Protocol

The osteointegration time was set at three months for the mandibular implants and six months for the maxillary implants. During the second-stage surgery, a mid-crestal mesiodistal flap was cut to ensure displacement of the keratinized tissue. The cover screw was manually unscrewed using a dedicated screwdriver and then replaced with a healing abutment. An initial impression was taken using alginate (Hydrogum5, Zhermack, Rovigo, Italy) to create an individualized impression tray. A second impression using polyether (Impregum^TM^, 3M Italia Srl, Milan, Italy) was then taken using the individualized open tray for a pick-up technique. Using a randomization table, the implants were assigned to either the PS group (example in Figure 2) or the NPS group (example in Figure 3) and the appropriate type of prosthodontics was then requested from the technician.

A resin screw-retained provisional prosthesis was delivered two weeks after the second-stage surgery. The provisional prosthesis was removed after three months and another impression was taken to manufacture the definitive metal-ceramic prosthesis according to the group assignment. All of the rehabilitations were cemented on milled standard abutments providing for a juxta-gingival emergence profile. The implants belonging to the PS group were rehabilitated with an internal conical 5° per side abutment, creating PS. Meanwhile, the rehabilitation for the NPS group used an internal flat-to-flat abutment connection with a matching platform (no PS) abutment. 

The occlusion was checked to remove pre-contacts and all interferences in centric, lateral, and protrusive movements. The definitive rehabilitations included both single crowns and partial fixed prostheses, up to a maximum of four elements.

### 2.4. Data Collection

A database was created to collect and process all of the implant and patient information. Patient anamnestic data were collected from the clinical exam and anamnesis. The analyzed variables were: sex, age, smoking status and the number of cigarettes/day (smoker: ≥10 cigarettes/day), systemic diseases, chronic or aggressive periodontitis, parafunction (bruxism, clenching), level of oral hygiene (0 to 4 score), and number of professional oral hygiene procedures in the last three years. 

The implant parameters analyzed at the time of the first surgery were: length, diameter, jaw position (maxilla/mandible/anterior/posterior), insertion torque, implant connection, and residual buccal bone width. Intraoral radiographs were used to evaluate the implants at the time of implant placement, when delivering the provisional and final crowns, and at the yearly follow-up appointments after the final rehabilitation. The implant depth (Figure 4) was determined based on a periapical radiograph at the time of the implant placement and recorded as the average distance from the implant shoulder to the first radiographic bone-to-implant contact (FBIC). 

After delivering the definitive crown, the presence of keratinized facial mucosa was checked and the prosthetic crown height (TH) was quantified (Figure 5). The TH was measured as the distance between the buccal gingival margin at the zenith and the crown incisal edge according to the main axis of the crown itself (Figure 5). This dimension was measured using a periodontal probe (UNC 15, University of North Carolina, Chapel Hill, NC, USA) with a resolution of 1 mm. When the incisal edge was not corresponding precisely to the probe’s measurement notches, the closer mark was chosen. In more detail, the TH was quantified directly in the mouth, observing the tooth perpendicularly to its facial surface. In the case of doubt, a second observer repeated the measure and an average dimension was registered. The baseline value was set at the time of the definitive prosthesis delivery, and the measurement was then repeated at one, two, and three years of follow-up from the final crown delivery.

### 2.5. Statistical Analysis

A randomized table was generated through a research randomizer (Redcap, Vanderbilt, Nashville, TN, USA) and followed to select the prosthetic protocol (PS or NPS) at the time of the impression. The requested sequence was one set of 90 balanced integer numbers with no sorting, ranging between 0 and 1, respectively associated with the NPS and the PS group. The assignment was consecutively associated with the implants, following the random set. A descriptive analysis was produced for the consecutively enrolled implants. The total follow-up time was three years from the time of the definitive prosthesis delivery, which occurred three to six months from the provisional prosthetic positioning. The TH baseline was determined at the time of the definitive crown delivery and the follow-up values were measured at one, two, and three years after. All data were then inserted into statistical software and processed (SPSS 20, IBM, New York, NY, USA). The implants were divided into two groups according to the presence of platform-switching (PS group) or no-platform-switching (NPS group). The TH was measured using deltas (ΔTH). The delta was measured by subtracting the follow-up TH value from the baseline TH. Consequently, positive values were associated with soft tissue recession, while negative values were related to coronal gingiva repositioning, referred to as definitive prosthodontic rehabilitation. Hence, recession was identified when the free buccal gingival margin measured at the zenith shifted apically. Meanwhile, gingival growth was associated with a coronal soft tissue migration of the gum. The means and statistical correlation refer to the ΔTH depending on the PS grouping. 

### 2.6. Sample Size Calculations and Correlation Analysis

The sample size was calculated (G*Power, Heinrich Heine Universität Düsseldorf, Düsseldorf, Germany) before the patient enrollment, based on two independent groups with a continuous distribution. As the distribution was supposed to be normal, a two sample two-tail *t*-test was applied. A sample size of 39 subjects per group was used in order to detect a 0.1 mm difference in the gingival height (ΔTH) and 0.15 mm standard deviation, with a required significance level between 0.05 and 80% power. Significance level and required power analysis were derived from previous literature [20,21,22,23,24,25]. At the end of the data gathering period, a Shapiro–Wilk test for normality was used to assess the sample distribution. Where the null hypotheses were rejected, the samples were evaluated as non-normally distributed and a Mann–Whitney test was applied to the independent samples. Platform-switching was set as the grouping variable: NPS was assigned to group 0 and PS to group 1, and TH and ΔTH at one, two and three years were defined as test variables. A Mann-Whitney test was performed at the baseline on the TH values to ensure randomization was successful, then to assess the ΔTH follow up significance levels and Z scores. Effect sizes were calculated using the equation $r=\frac{Z}{\sqrt{N}}$. The resulting effect sizes were converted into Cohen’s effect sizes for simplicity and ease of use. The same procedure was applied to the ΔFBIC values. 

## 3. Results

A total of 90 implants were placed according to the standardized surgical and prosthetic protocols fixed at the beginning of the prospective study. 

Among the 90 implants enrolled in the study, 77 implants were finally considered in the statistical analysis. The drop-out implants were: two due to missing radiographs during the treatment; nine belonging to patients who were unable to attend the follow-up, were no longer contactable, or moved to a different city; and two that presented signs of mucositis during the observation period. All of the implants (AnyRidge, MegaGen, Seoul, Korea) provided a flat-to-flat connection with the same-brand abutment and were placed in both the anterior and posterior region, and the upper and lower jaw. The implant distribution is reported in Table 1 and Table 2. 

The FBIC mean for the evaluated implants at the time of placement was −0.28 ± 0.67 mm. However, at the three-year follow-up, the FBIC mean was 0.66 ± 0.97 mm for the NPS group and −0.05 ± 0.91 mm for the PS group (Table 3). As for the ΔFBIC, correlation was observed during the first two years of follow up. The first year showed *p* ≤ 0.016 (mean 0.237 mm, S.D. = 0.804 mm for the PS group and a mean 0.721 mm, S.D. = 1.050 for the NPS) and a Cohen’s *d* = 0.517, C.I. = 0.063–0.971. In the second year, significance dropped (*p* = 0.019, mean = 0.229 ± 0.932 and 0.777 ± 1.068 for PS and NPS respectively), while Cohen’s d was 0.546, C.I. = 0.091, 1.001. In the third year, significance for the ΔFBIC was weak or absent. The group distribution was: 39 implants (50.6%) in the NPS group and 38 implants (50.4%) in the PS group. After three years, a mean recession (ΔTH) of 0.54 ± 1.93 mm was recorded for the NPS group, whereas a mean coronal migration of 0.17 ± 0.91 was recorded for the PS group (Table 4). In more detail, no correlation was found at the one-year follow-up *p* = 0.090. However, the two-year (*p* = 0.015, mean = −0.11 mm, S.D. = 0.84 mm, *d* = −0.62, C.I = −1.08–0.16) and three-year (*p* = 0.031, mean = −0.17 mm, S.D = 0.95 mm, *d* = −0.59, C.I. = −1.05–0.13) follow-up data indicated significantly lower gingival recession in the PS group compared to the NPS group.

At the time of data analysis, the randomization distribution of NPS and PS had not balanced, anymore due to the dropouts, thus 39 NPS and 38 PS were collected. 

The calculated effect sizes (Cohen’s *d*) also confirmed the clinical relevance of these findings. Due to the drop-out rate, the power achieved for the present sample study was 76% for the two-year follow-up and 70% for the three-year follow-up. Consequently, these outcomes confirm that the absence or presence of platform-switching would appear to affect the tendency of the gingival buccal margin towards recession or creeping over time (Table 4) (Figure 6). Creeping has been defined as coronal growth of the gum margin in the absence of any inflammatory sign [25].

The effect of the abutment platform-switching seemed to reach a plateau around the second year after prosthesis positioning, as shown in Table 3. Moreover, a greater marginal bone loss was observed in the NPS group over the follow-up period (Table 3) (Figure 7).

## 4. Discussion

Many clinical investigations and systematic reviews have already shown that platform-switching at the implant-abutment interface significantly reduces crestal bone loss in comparison with a regular flat-to-flat connection [26], which has led to the increased use of platform-switching in regular clinical implantology. Several studies have also reported reduced bone resorption with platform-switched implants when compared with platform-matched implants [27]. From a technical point of view, the switching platform preserves the crestal bone by providing a horizontal biological width formation over the implant neck [28] and shifting the implant-abutment microgap distant from the bone crest [29]. The microgap is one of the major factors related to apical bone remodeling [30]. Yet, despite extensive data on soft tissue stability around platform-switched implants [31,32,33,34,35], there is still limited comparative information of implants with and without PS. In the present study, that followed 77 dental implants for three years from prosthetic delivery, apical repositioning of the soft tissue was observed in both the PS and NPS groups during the first year. However, the second year showed divergent behavior between the two groups, where the PS group showed a reduced vertical crown dimension due to soft tissue creeping, while the NPS group exhibited soft tissue apical recession. Moreover, after three years, the PS group showed increased soft tissue creeping values.

Another important factor influencing bone remodeling around PS implants is the soft tissue thickness at the time of implant placement [29]. The behavior of PS implants with thin (<2 mm) and thick (>2 mm) soft tissue was previously analyzed by Linkevicius, where thin peri-implant soft tissue at the time of implant placement did not prevent bone loss, while thick tissue maintained the crestal bone level with minimal remodeling [29]. Therefore, with a thin biotype, platform-switching may be insufficient to prevent soft tissue recession over time around a dental implant. In this case, soft tissue augmentation is needed at the time of the implant placement [29].

An experimental study on humans showed that at eight weeks, the soft tissue consists of a barrier epithelium of 1.9 mm and a connective tissue portion of 1.7 mm [36,37]. This suggests that the bone undergoes remodeling to create sufficient space for a peri-implant seal to be formed [36]. 

One of the roles of the connective tissue zone is to support the epithelial tissue and limit its migration towards the apical direction [31]. PS implants facilitate the formation of a connective tissue ring over the implant shoulder [32], creating thicker connective tissue that can provide better protection of the surrounding bone and reduce bone remodeling in the apical direction [28].

Therefore, the present study can verify the reduction of recession risk with PS implants. The formation of a connective ring facilitated by a PS implant may result in thicker connective tissue [36], which in turn may prevent bone remodeling [28]. This bone stability may then result in better tissue stability, and a thicker connective tissue has the potential to creep [38]. 

Notwithstanding, the present study has several limitations, as the results may have been influenced by different confounding factors and potential interactions, which were not investigated. The positions of the implants in the mouth (maxilla, mandible, anterior, and posterior) may be conditioned by the specific phenotype, resulting in different tissue behavior at the maturation pattern and during the establishing of the biologic width. In addition, the crestal soft tissue thickness at the time of the implant placement may induce a bone remodeling, according to Linkevicius [29], when the thickness is below 2mm, thus, again, influencing the tissue adaptation. A diet diary may also be introduced to check the avoidance of food repercussions on healing. In addition, the manual measurement of the TH with a probe has a limited resolution. Hopefully, such studies may be conducted in the future as follow: with TH measurement through digital measurement evaluated by an independent observer; after a digital impression or a standardized picture, a minimum threshold should be set for crestal soft tissue thickness; and oral sectors should be analyzed independently.

With regards to the esthetic outcome, interproximal soft tissues should be monitored in future protocols.

## 5. Conclusions

Within the limitations of the present research, it can be concluded that the absence or presence of platform-switching would appear to affect the tendency of the gingival buccal margin towards recession or creeping. Implant–abutment platform-switching also seems to prevent peri-implant soft tissue recession over time when compared to implants with no dimension switching of the abutment.

Furthermore, the platform-switching effect seemed to reach a plateau around the second year after prosthesis positioning. 

## Figures and Tables

**Figure 1 jcm-10-02955-f001:**
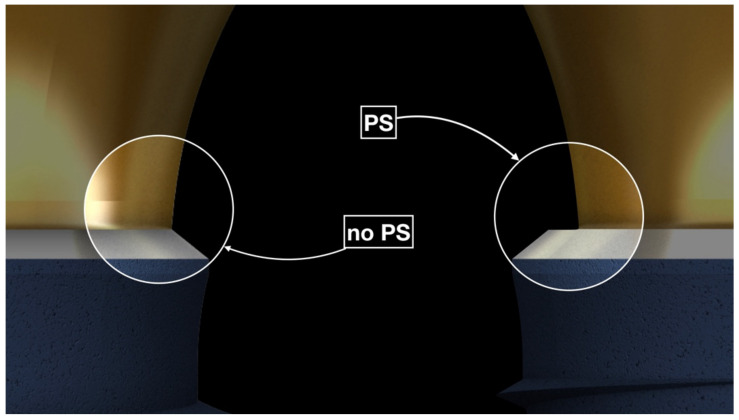
Two implant designs: with platform-switching (PS) technology and without. It can be noticed that the switched abutment is narrower than the implant platform.

**Figure 2 jcm-10-02955-f002:**
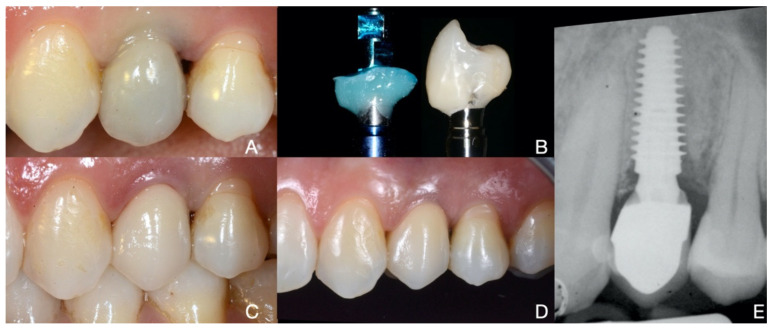
A provisional screw-retained crown (**A**) is set in place in Section 2.4. After three months, the emergence profile is transferred to the technician through the individualized transfer technique (**B**) to reproduce the same emergence profile of the provisional in the final crown. The PS final rehabilitation (**C**) is delivered. After three years the crown shows acceptable integration (**D**) and good stability of the buccal gingival zenith. On the right (**E**) the X-ray is acquired at the time of the final crown delivery.

**Figure 3 jcm-10-02955-f003:**
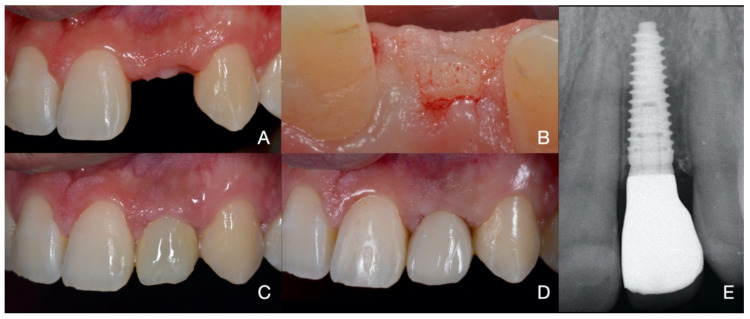
In the edentulous ridge (**A**) is placed an implant in Section 2.2. Six months later, the second stage surgery (**B**) is performed through a mid-crestal flap for the minimal displacement of the keratinized tissue and a provisional screw-retained crown (**C**) is set in place. Three months later the NPS final rehabilitation is delivered. (**D**). On the right (**E**) is the three years X-ray.

**Figure 4 jcm-10-02955-f004:**
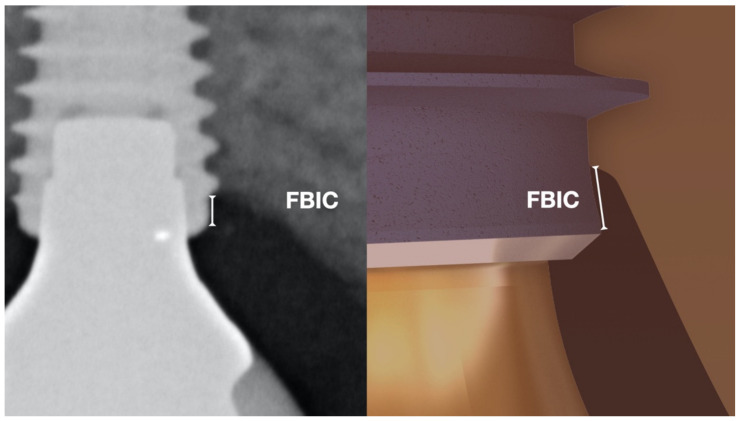
FBIC (first bone to implant contact) measure on a periapical radiograph (**left**) and the schematic detail (**right**).

**Figure 5 jcm-10-02955-f005:**
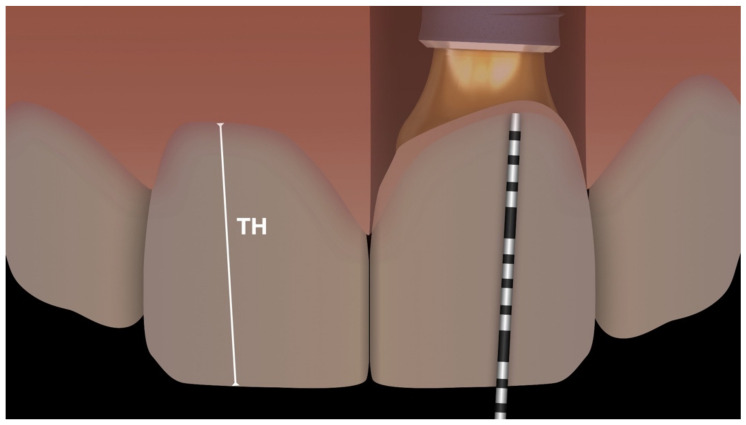
TH measurement through the probe. tooth height (TH) measured as the distance between the buccal gingival zenith and the crown incisal edge, according to the main axis of the crown.

**Figure 6 jcm-10-02955-f006:**
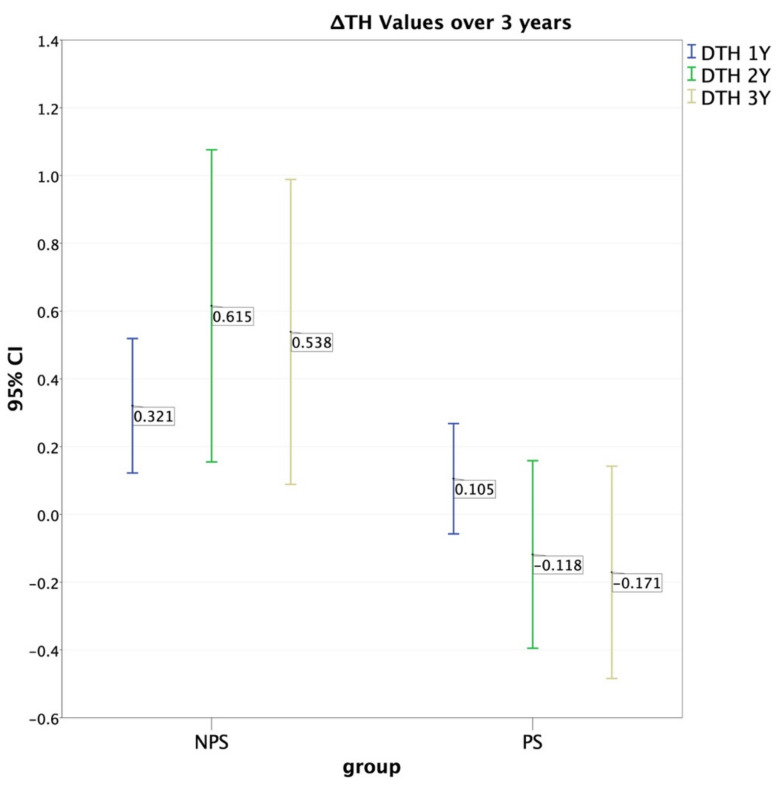
ΔTH during follow-up period: NPS implants exhibited greater recession when compared to PS implants. Negative values (mm) mean creeping or coronal migration of gingival margin, positive values mean recession.

**Figure 7 jcm-10-02955-f007:**
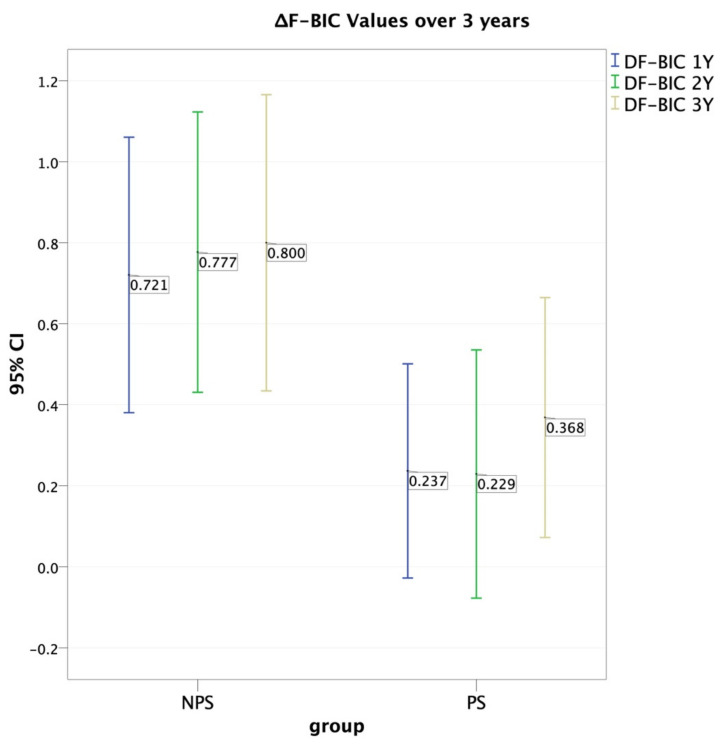
ΔFBIC values during follow-up period. Amount of bone loss (mm) was higher for NPS implants compared to PS implants during observation period.

**Table 1 jcm-10-02955-t001:** Implant distribution considering implant site and features.

Position	Abutment	Upper Jaw	Lower Jaw	Total
Anterior	PS	4	-	4
	No PS	5	3	8
Posterior	PS	14	20	34
	No PS	9	22	31
Total		32	45	

**Table 2 jcm-10-02955-t002:** Implant distribution considering sex and age.

Sex	Abutment	N. Patients	Mean Age
Female	PS	25	62 ± 13
	No PS	26	64 ± 7
Male	PS	13	62 ± 13
	No PS	13	62 ± 7
Total		77	63 ± 11

**Table 3 jcm-10-02955-t003:** FBIC and ΔFBIC values during follow-up period. Amount of bone loss (mm) was higher for NPS implants compared to PS implants during observation period.

			ΔFBIC Values and Correlations for PS and NPS Implants																
		Platform-Switching (PS)							No Platform-Switching (NPS)						Student T Significance		Bonferroni’s Alpha	Cohen’s *d* Effect Size	95% Confidence Intervals for Cohen’s *d*
		n			Mean		S.D.		n		Mean		S.D.						
ΔFBIC 1Y		38			0.237		0.804		39		0.721		1.050		*p* = 0.016		*p* = 0.048	0.517	0.063, 0.971
ΔFBIC 2Y		38			0.229		0.932		39		0.777		1.068		*p* = 0.019		*p* = 0.057	0.546	0.091, 1.001
ΔFBIC 3Y		38			0.368		0.901		39		0.800		1.129		*p* = 0.068		*p* = 0.204	0.422	−0.03, 0.874
			**FBIC Values and Correlations for PS and NPS Implants**																
				**Switching (PS)**						**Non Switching (NPS)**						**Mann Whitney** **Significance**	**Bonferroni’s Alpha**	**Cohen’s *d* Effect Size**	**95% Confidence Intervals for Cohen’s *d***
				**n**		**Mean**		**S.D.**		**n**		**Mean**		**S.D.**					
	FBIC 1Y			38		−0.179		0.633		39		0.577		0.836		*p* ≤ 0.001	*p* ≤ 0.003	1.018	0.544, 1.493
	FBIC 2Y			38		−0.187		0.879		39		0.633		0.924		*p* ≤ 0.001	*p* ≤ 0.003	0.909	0.44, 1.378
	FBIC 3Y			38		−0.047		0.914		39		0.656		0.970		*p* = 0.003	*p* = 0.009	0.747	0.285, 1.209

**Table 4 jcm-10-02955-t004:** Correlation between prosthetic crown height variation (ΔTH) and presence or absence of platform-switching at one, two, and three years. Difference becomes significant (Mann–Whitney, 95% conf.) at two and three years from delivery of final crown.

	Baseline TH & ΔTH Values and Correlations for PS and NPS Implants										
		Platform-Switching (PS)			No Platform-Switching (NPS)			Mann- Whitney Significance	Bonferroni’s Alpha	Cohen’s *d* Effect Size	95% Confidence Intervals for Cohen’s *d*
		n	Mean	S.D.	n	Mean	S.D.				
Baseline TH		38	7.859	2.155	39	7.184	1.574	*p* = 0.097	−	−	−
ΔTH 1Y		38	0.105	0.495	39	0.320	0.612	*p* = 0.09	*p* = 0.27	*d* = −0.764	−1.226,−0.30
ΔTH 2Y		38	−0.118	0.842	39	0.615	1.421	*p* = 0.015	*p* = 0.045	*d* = −0.626	−1.084,−0.169
ΔTH 3Y		38	−0.171	0.953	39	0.538	1.388	*p* = 0.031	*p* = 0.093	*d* = −0.594	−1.051,−0.138

## Data Availability

Data supporting the results obtained can be requested to the corresponding author (mattiamanfredinidr@gmail.com) or the person responsible for the statistical analysis (pm.pasini@gmail.com).

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
