# Peer review of "Behavior of Soft Tissue around Platform-Switched Implants and Non-Platform-Switched Implants: A Comparative Three-Year Clinical Study"

_jcm, 2021, doi:10.3390/jcm10132955_

Round 1

Reviewer 1 Report

Dear authors,

It was a pleasure to review your manuscript which is well written and rich in significant clinical contents. 

There are few minor corrections that should be followed before proceeding with publication.

Please consider rewriting the paragraph included between line 45 and 49 which is a little unclear.

Despite being considerate to be mentioned, figure 2 would have been more useful as an actual radiograph. This is pretty cartooned and forced. 

Furthermore, did you measure TH directly in the mouth? According to literature, esthetic parameters on implants are usually assessed on standardized intraoral photographs by independent observers. However, please provide further details on your method. It would have been relevant mentioning the outcome of soft tissue at interdental level.

In the statistics methodology, you mentioned the sample size calculation being referred to “A sample size of 39 subjects per group was used in order to detect a 0,1 mm difference in the gingival height and 0,15 mm standard deviation, with a required significance level of 0.05 and 80% power”

Please mention if this difference is clinically significant according to you or according to previous literature.

Since you measured sample size on TH, please declare it as your primary outcome or at least as the most demanding one with regards to its assessing instruments accuracy.

In the methods you mention a Correlation Analysis. Please describe how you performed it.

Tables and graphics are very clear, but you should mention numbers also in the text of the Results section.

Again, inter proximal soft tissue outcome report would have been appreciated as the esthetics of implant highly depends on distal and mesial papilla representation.

Since the focus of the manuscript is on implant esthetics, please provide the photographs of at least one clinical case per group.

Author Response

Dear Editor and Reviewers,

the response point by point has been attached.

Thank you for your time and your attention.

Reviewer 2 Report

I am not sure that the small difference is going to change the treatment. I believe that platform-switching is a proven decision and since Rich Lazzara introduced this thought. There are many publications missing from the references.

Author Response

(The authors gave the same response as above.)

Reviewer 3 Report

  1. Cone-Beam TC should be corrected to Cone-Beam CT. (line 110).
  2. Alveolar bone width on the facial and lingual of implants at the time of implant insertion were not mentioned. This alveolar bone on the facial of implants might contribute to gingival recession. 

Author Response

(The authors gave the same response as above.)

Reviewer 4 Report

This manuscript sought to verify the influence of platform switching (PS) on soft tissue behavior by comparing the soft tissue stability around implants with and without PS during 3 years of follow-up. This is an interesting manuscript but I am concerned regarding the basic nature of the statistical analysis performed and the use of multiple testing for the same patients over different time points. Why didn’t the authors use longitudinal analysis, either with transformations of the outcome or non-parametric analysis. The author’s statistical analysis did not control for potential confounders (e.g. sex, age at baseline) not investigate potential interactions. This manuscript may serve to be an initial investigation but should not claim to “provide a biological explanation for the absence of recession with PS implants”.

This manuscript also requires correct formatting (e.g. Tables with inconsistent use of decimals and wrapping numbers).

Specific comments are detailed below:

  • The introduction reads well but would benefit from breaking the content up into three or four paragraphs
  • Figure 1 – Please explain the differences in with/without platform switching (PS) in this diagram as the difference between the two is not clear to the untrained eye
  • LINES 91 to 93: Please clarify the data source for the 178 patients
  • LINE 96 – Please amend from “age ≥18” to “age ≥18 years”
  • The Surgical protocol paragraph is very detailed and would be better broken up into clearly defined separate paragraphs or a flowchart of the procedures may be better so that the reader can easily comprehend this protocol.
  • LINE 135 – How did you ensure the all patients observed a 10-day liquid diet plus 15 days of soft foods following implant placement.? Did they keep a diary of what they consumed? Some patients may have consumed sweet soft foods whilst others did not and could this have influenced the results. Please clarify this.
  • LINES 151 to 153 – Please outline if the randomization table result did/did not result in balanced groups and what are the numbers for the types of prosthodontics in each group.
  • LINES 223 to 224 – Was the Mann-Whitney test applied to the independent samples at baseline?
  • Table 1 – Please add total rows and columns.
  • LINE 249 – This should be Table 2. Please format correctly as the figures wrap around. You have performed multiple tests at three time points. Did you correct for multiple comparisons (e.g. Bonferroni adjusted alpha)? If so, please include this in your methods section. Please be consistent with the use of the decimal place – you have some figures using ‘,’ and others using ‘.’. You have a Cohen’s d effect size > 1 -
  • LINE 260, 269 – Please rename “Graphic” as “Figure”. Also, these figures should show a confidence interval rather than point values.
  • LINE 262 – Please ensure you format this table so the values do not wrap around
  • LINES 307 to 308 – Your statement “Therefore, the present study can provide a biological explanation for the absence of recession with PS implants.” is a very challenging as I do not believe you consider many biological confounding factors in your analysis.
  • LINES 312 to 314 – Please expand on your limitations section in your Discussion regarding how the position of the implant in the mouth (maxilla, mandible, anterior, posterior) and the soft tissue thickness at the time of implant placement would impact on your results. Also, there are further limitations you have not indicated the limitation (e.g. nature of nature of food content/diet).
  • Please check you reference tables and figures in your text prior to their inclusion

Author Response

(The authors gave the same response as above.)

Round 2

Reviewer 1 Report

Dear authors,

thank you for your reply, I will recommend acceptance of your study.

Reviewer 3 Report

Well done
